# Recent Advances in Sources, Migration, Public Health, and Surveillance of Bisphenol A and Its Structural Analogs in Canned Foods

**DOI:** 10.3390/foods12101989

**Published:** 2023-05-14

**Authors:** Ling Ni, Jian Zhong, Hai Chi, Na Lin, Zhidong Liu

**Affiliations:** 1East China Sea Fisheries Research Institute, Chinese Academy of Fishery Sciences, Shanghai 200090, China; ni.lu.yi2006@163.com (L.N.); chih@ecsf.ac.cn (H.C.); lina903368043@163.com (N.L.); 2Shanghai Key Laboratory of Pediatric Gastroenterology & Nutrition, Shanghai Institute for Pediatric Research, Shanghai Jiao Tong University School of Medicine, Shanghai 200092, China; jzhong@shsmu.edu.cn

**Keywords:** bisphenol A, structural analogs, canned foods, sources, public health, surveillance

## Abstract

The occurrence of bisphenol A (BPA) and its structural analogs, known as endocrine disruptors is widely reported. Consumers could be exposed to these chemicals through canned foods, leading to health risks. Considerable advances have occurred in the pathogenic mechanism, migration law, and analytical methodologies for these compounds in canned foods. However, the confusion and controversies on sources, migration, and health impacts have plagued researchers. This review aimed to provide insights and perspectives on sources, migration, effects on human health, and surveillance of these chemicals in canned food products. Current trends in the determination of BPA and its structural analogs have focused on mass spectroscopy and electrochemical sensor techniques. Several factors, including pH, time, temperature, and volume of the headspace in canned foods, could affect the migration of the chemicals. Moreover, it is necessary to quantify the proportion of them originating from the can material used in canned product manufacturing. In addition, adverse reaction research about exposure to low doses and combined exposure with other food contaminants will be required. We strongly believe that the information presented in this paper will assist in highlighting the research needs on these chemicals in canned foods for future risk evaluations.

## 1. Introduction

Bisphenol A (BPA) was first reported in 1891 by the Russian chemist Aleksandr Dianin and was produced via the condensation of acetone with phenol in 1905 [1]. BPA was subsequently found in the production of food containers, preventing food from directly reaching the metal surface.

Studies have presented BPA as a well-known endocrine disruptor regularly detected in biological samples such as amniotic fluid, blood, breast milk, urine of mammalian, etc. [2,3]. The contamination level of BPA in canned foods is frequently higher than that in non-canned foods (e.g., fresh and frozen) [4]. Dietary exposure through canned foods accounted for about 90% of the total interior exposure (conjugated plus unconjugated) to BPA based on central total exposure calculations [5]. A study measured the concentration of BPA in convenience foodstuffs from Dallas, Texas, USA, and reported that BPA was present in 73% of canned foods and 7% of non-canned products. Moreover, the dietary exposure to BPA was calculated to be 12.6 ng/kg bw/day, of which 12.4 ng/kg bw/day came from canned foods [6]. The effects of BPA as a xenoestrogen are associated with several diseases, such as adverse neurological diseases [7], immunological diseases [7], diabetes [8], obesity [9], cardiovascular diseases [10], renal diseases [11], behavior disorders [12], cancer [13], and reproductive disorders [14]. 

Accordingly, the European Chemicals Agency put BPA on the candidate list as a substance of very high concern in January 2018 [15]. Moreover, recent regulations further restrict the use of BPA in food contact materials (FCMs) [16]. Furthermore, the French law No 1442/2012 was aimed at prohibiting the manufacture, export, import, and commercial use of all food packaging materials containing BPA [17].

The production and consumption of BPA have been reduced in many countries, and replacement with bisphenol analogs (Table 1) is actively being explored in several manufacturing industries to produce “BPA-free” products [18]. In recent years, increasing concentrations of BPAF, BPF, and BPS similar to or even higher than that of BPA have been reported in the environment and human urine in some regions [19]. Thus, exposure in human life to BPA analogs mainly occurs through dietary intake via foods and FCMs [20]. The change in lifestyle has increased the demand for ready-to-eat and canned foods [20]. Furthermore, consumer behavior shifted during the COVID-19 pandemic, with the demand for canned foods significantly increasing [21]. Unfortunately, the transfer of these chemicals from FCMs to food contents has been reported in canned foodstuffs and soft drinks [22]. Therefore, we analyzed research trends concerning BPA analogs in canned food products based on 62 references from the Web of Science database. Figure 1A gives the annual publication numbers from 2008 to 2022. Annual publication numbers experienced continuous growth generally and reached a peak in 2020. Figure 1B illustrates countries contributing to research on BPA analogs in canned foodstuffs. Among these countries, China and Spain have made the greatest contribution to the research on BPA. 

Only a small number of studies are concerned with the hormonal action of BPA analogs, and the majority of these studies show that the analogs have similar health issues as BPA [18]. Furthermore, some BPA substitutes seem to have higher estrogenic effects than BPA, according to in vivo and in vitro assays [7,23]. It is painfully apparent that the usage of these chemicals is rising globally [24]. 

Previous reviews mainly focused on the properties of BPA, dietary exposure sources, detection methods, and toxicity of BPA. On the other hand, reviews covering additional sources of exposure (such as canned food contact materials), migration law, and simultaneous monitoring of BPA and its analogs in canned foodstuffs are still lacking [25]. The present review briefly introduces the sources and influencing factors of BPA migration concerning canned foods. Furthermore, it provides an overview of the current state of knowledge on human exposure and the relevant toxic mechanism of these compounds. Moreover, updates on the analytical methods used for the identification and legislation of the BPA substitutes in FCMs worldwide are also presented as a graphical abstract. We believe that the information in this review will help identify emerging trends of control on BPA and BPA analogs exposure and establish regulations according to their categories rather than their individual compound.

## 2. Contamination Sources of BPA and Its Analogs in Canned Foods

The exposure of BPA and its analogs from canned food products to humans is mainly as a result of their migration from FCMs, such as polycarbonate (PC) plastic and epoxy resin [15]. Another possibility to consider is raw materials contamination by BPA and its analogs. Deceuninck et al. suggest that BPA contamination of meat is obtained through contact with material containing BPA (e.g., during processing) [26]. 

### 2.1. BPA and Its Analogs from Packaging Materials Used in Canned Foods

The inner surface of the metallic can is coated with a polymer film to prevent food corrosion and contamination. There is strong evidence indicating that the polymerization of the lacquer may be incomplete, causing a large number of unreacted compounds to be released into the foods [27]. For instance, a study reported the migration of these chemicals from coated tin cans to vegetable foods widely consumed worldwide [28]. Another study reported the migration of the chemicals from the inner paint of tinplate cans (four brands) to foodstuff simulants [29]. Cao et al. verified that migration from can coatings is the likely source of BPA in canned products [30]. Furthermore, the European Food Safety Agency (EFSA) conducted a dietary BPA exposure assessment in a scientific opinion published in 2015. The report indicated that the highest concentrations of BPA were detected in packaged products (18.68 μg/kg) compared to unpackaged foods (1.5 μg/kg) [31].

### 2.2. BPA and Its Analogs from Raw Materials of Canned Products

BPA and its analogs migration from the can coating is likely to be the principal source of these chemicals in canned foods with relatively high concentrations. However, the origin of the raw materials plays a crucial role in the exposure of humans to these chemicals. In the scientific opinion published by EFSA in 2015, the highest concentrations of BPA were detected in fish and meat, with values of 7.4 μg/kg and 9.4 μg/kg, respectively [31]. 

Several studies have confirmed the above results. A study by Akhbarizadeh et al. reported that BPA was found in almost all the samples, with BPB being the second most compound in concentration [32]. Moreover, the higher concentration of these compounds in top predators of consuming benthic organisms indicated that food habits affect interspecies distribution. The direct contamination of BPA in aquatic products could be caused by the leaching of microplastics in the marine environment and organisms [33]. 

It should be noted that the researcher from the previous study analyzed liver samples obtained from pregnant ewes with continuous subcutaneous exposure to BPA for 105 days and identified plausible findings indicating observed contaminations resulted from direct contact with BPA-containing materials during meat processing and bioaccumulation in animals used as raw materials for canned foods via migration from plastic packaging into the solid feed [34]. According to Zhou et al., the prevalence of BPA and BPS determination in fresh vegetables from Zhejiang Province from June 2017 to April 2018 was 75.0% (18/24) and 62.5% (15/24), respectively, with levels of up to 4.5 and 3.2 μg/kg [35]. BPA was also found in unpacked apples and pears, resulting from contaminations during primary production [22]. In a separate study, BPS was not found in 159 different canned food samples from Quebec City, Canada, but it was found in 9 meat-based food samples (1.2–35 ng/g), indicating BPS contamination can come from sources other than can coatings [36].

## 3. Factors Influencing the Migration of BPA and Its Analogs in Canned Foods

Research has suggested that BPA and its analogs can migrate from coating containing PC to food in cans through two distinct pathways, namely, the diffusion of residual chemicals in PC after the manufacture and hydrolysis of the coating polymer [37]. Previous studies suggested that the variability in the concentration of these compounds in canned foods may arise from several reasons. El Moussawi et al. reported that the variations could be from variations in the elution rates (Figure 2), the composition of epoxy can coatings, canned foods processing (heating process), and storage conditions (pH, time, temperature, and volume of headspace of canned foods) [29]. Moreover, factors such as fat content, the presence of oxidizing agents and nitrates, and packing media (layer, coating thickness, and particle sizes) also contribute to the variability in the concentration of BPA and its analogs [38]. In general, the migration of BPA from coating to food could be promoted with increased storage temperature and time, and initial concentrations [29,33,38,39]. In addition, the initial concentration of the compounds in the coating of FCMs affects the extent of migration [33]. For instance, Zhang et al. investigated four epoxy coatings (three of the cans were made from aluminum and one from steel) and found BPA, BPC, and BPF in the aluminum cans [38]. Moreover, Wagner et al. demonstrated that the apparent diffusion coefficients of BADGE and BADGE·HO were positively correlated with film thickness and negatively correlated with crosslink density [39].

Furthermore, Kang et al. reported that the effect of heating temperature on BPA migration is more significant than heating time [40]. Additionally, Choi et al. estimated the contamination levels of BPA, BPB, BPF, and BADGE·2HCl in canned foods and indicated that concentrations of the compounds migration into canned foods vary greatly. The relatively lower concentrations in canned fruits and vegetables are more likely due to the hydrophilic properties of these canned foods rather than an interaction with the canned lining [41].

## 4. Surveillance of BPA and Its Analogs in Canned Foods

The effect of BPA and its analogs on human health is still controversial. Therefore, their concentration in canned foods is considered to be determined through an extensive monitoring program to establish a possible causal relationship of exposure risk.

### 4.1. Analytical Methods for Determining BPA and Its Analogs in Canned Foods

In the past decade, significant progress has been made in the analytical methodologies for these chemicals in canned foods. In this review, 99 publications about analytical techniques for these compounds published since 2013 were obtained from the Web of Science using the keywords “bisphenol”, “canned foods”, and “risk assessment” or “determination” on 30 September 2022 (Figure 3). The data indicate conclusively that the mass spectrum (MS) detector combined with both high-performance liquid chromatography (HPLC) and gas chromatography (GC) is the most common approach. The analysis of LC performs without derivatization steps, resulting in shorter sample preparation time and better quantification limit. In contrast, the GC technique implied that a derivatization process can simultaneously determine all studied compounds with sufficient repeatability and accuracy [42]. Moreover, an enzyme-linked immunosorbent assay (ELISA) technique was established for the determination of BPA and its analogs in several studies [43]. The main advantages of these detectors include their high selectivity, high sensitivity, and universality [42]. However, these methods have some drawbacks, including requiring professional analysts, high costs, and the inability to conduct on-site analysis. Simplified analytical procedures are an attractive strategy for detecting small molecules. Accordingly, considering its high toxicity even at trace levels, an electrochemical sensor has been developed to detect these compounds in a fast, portable, highly sensitive, and cost-efficient manner in recent years [44,45,46,47]. The development of electrochemical sensors shows an increasing trend in Figure 3. Interestingly, the proportion of electrochemical sensors in all retrieved detection methods in 2022 was around 25.0%.

### 4.2. Estimates of BPA and Its Analogs in Canned Foods

The presence of these compounds in canned meat, fish, fruits, and vegetables has been reported frequently. Table 2 compiles the residual levels of BPA and its analogs related to different canned products.

Some studies present that the detected concentration of various compounds varies depending on the different compounds in them and the origins of the canned food. For instance, BPA concentration in canned foods (ND to 837 μg/kg) was higher than BPF (ND to 75.4 μg/kg) and BPS (ND to 1.6 μg/kg) [48]. In addition, González et al. compared BPA, BPB, and BPE levels in canned foods from Spain and reported that BPA was significantly higher than BPB and BPE in canned foods [22]. Moreover, the concentration of BPA in canned foods from Asian countries (e.g., China and Korea), except Thailand, was higher than that in European countries (e.g., Italy, Spain, Portugal, and Greek) [22,49,50,51,52]. Canned vegetables and grains collected in Spain and Lebanon were among the food categories with lower levels of BPA and its analogs [27,52].

## 5. Impact of BPA and Its Analogs from Canned Foods on Public Health

Exposure to BPA and its analogs through diet intake is an issue of great concern. Exposure through dietary sources affects many people and may appear in trace amounts over a long period of time without being detected [57]. Lucarini et al. described the exposure of infants and toddlers to BPA and 14 emerging BPA analogs (i.e., BPAF, BPAP, BPB, BPC, BPE, BPF, BPG, BPM, BPP, BPPH, BPS, BPTMC, and BPZ), and found that these compounds were present in the urine of 47% of infants and toddlers [58]. Therefore, the data on exposure assessments and potential toxicity mechanisms for these contaminants in canned foods are essential for public health monitoring.

### 5.1. Exposure Assessment Examples of BPA and Its Analogs by Analysing Residual Levels of These Chemicals in Canned Foods

Exposure assessment of ingesting BPA and its analogs through the consumption of canned foods (Table 3) indicated health concerns for consumers in several countries, including Spain [52], China [48], the United States of America [6], France [50], and South Korea [41]. The average exposure levels of Chinese consumers to BPA from canned products were estimated to be 32.9 ng/kg bw/day [48]. This exposure was slightly higher than that in the United States of America (12.6 ng/kg bw/day), France (22.5 ng/kg bw/day), South Korea (19.33 ng/kg bw/day), and Spain (21.5 ng/kg bw/day) [52]. The exposure reported for Spain was significantly lower than the 370 ng/kg bw/day [22]. Moreover, the average exposure levels to BPA analogs through dietary intake in Spain (156.5 ng/kg bw/day) [52] was significantly higher than that in China (5.3 ng/kg bw/day) [48], Korea (25 ng/kg bw/day) [50], and France (17.15 ng/kg bw/day) [50]. Exposure to the previously mentioned compounds below the limit of tolerable daily intake (TDI) (<4 µg/kg bw/day) set in 2015 by EFSA indicated dietary intake from canned foods poses low risks to the general population, even with combined exposure levels of different categories. However, EFSA issued a draft to further reduce the TDI of BPA on 15 December 2021, from 4 to 0.04 µg/kg bw/day [31]. Furthermore, a significant effect may not be observed when each chemical is present at a low dose. Therefore, combined exposure levels should be further considered.

### 5.2. Toxicological Mechanism of BPA and Its Analogs in Canned Products

These chemicals have received special focus in the past few years mainly due to their diffused presence in foods and their toxicological effects on human health. BPA, as a lipophilic-synthetic-organic compound, is orally absorbed through the gastrointestinal tract and transported to the liver, where it is metabolized and acquires hydrophilic characteristics [59] and is excreted via the bile and urine in unconjugated and conjugated form (BPA-glucuronide or BPA-sulfate) with a half-life of approximately 6 h [14]. Nevertheless, the expression of the β-glucuronidase enzyme separates the BPA-glucuronide group from the metabolite through hydrolysis, then distributes around the body through the deconjugation of BPA and releases its active form into the blood [60].

Given the analogs’ molecular structural similarity to BPA, they have a similar endocrine-disrupting mechanism and potential to BPA [61,62,63]. For instance, three key pathways: “Necroptosis”, “Adipocytokine signaling pathway”, and “C-type lectin receptor signaling pathway” were observed in zebrafish following BPF and BPS exposure in a study [64]. Surprisingly, all three pathways were closely related to the potential risk of cancer [64].

The toxicological mechanisms of these compounds were studied in detail by several research groups over the years (Table 4). Available scientific data indicate that BPA could cause changes in promoter methylation of tumor suppressor genes [65,66]. Meanwhile, these chemicals induce hyperglycemia by impairing insulin signaling transduction and disrupting glucose metabolism, increasing the risk of non-alcoholic fatty liver disease in male zebrafish [67], and stimulating the expression of autophagy and inflammatory genes, increasing the levels of triacylglycerol and total cholesterol in male zebrafish liver, and even inducing fibrosis and hepatic apoptosis resulting in nonalcoholic steatohepatitis [68]. Moreover, exposure to these chemicals promotes the development of obesity through multiple pathways, including (1) upregulating the expression of a liver lipogenic enzyme which has great obesogenic effects [69]; (2) suppressing the mRNA expression of gene encoding insulin leading to poor insulin production, which results in obesity via inducing hyperglycemia [70]; (3) disrupting triacylglycerol metabolism resulting in obesity [71]; (4) significantly reducing the mRNA levels of lipogenic transcription factor srebp1 and increasing that of fatty acid synthetase [72]. In addition, previous research demonstrated that these compounds could induce intracellular Ca^2+^ homeostasis disturbance [73] and downregulated the expressions of neurodevelopment-related genes, leading to neurotoxicity [70,74].

Rochester et al. evaluated the effects of BPF and BPS on the physiological and endocrine activities and compared their hormonal efficacy with that of BPA. Interestingly, the result revealed that BPS and BPF also exhibited other effects in vitro and in vivo, such as changing organ weights, reproductive endpoints, and enzyme expression [61].

Of note, a sustained discussion on hazard identification and characterization has continued over a number of years. The uncertainty is not only related to concerns about the degree of adverse health effects but also includes the dose levels causing health problems. EFSA has demonstrated that BPA is a reproductive substance at a high dose, but its effect is doubtful at a lower dose [31]. There is some confusion regarding the scope of “low-dose”. Furthermore, the low dose levels could cover approximately 8–12 orders of magnitude. However, most conducted doses in vivo animal and in vitro cellular studies far exceed the dose range of human exposure [4].

## 6. Bisphenols Current Legislation

The European Commission has published Regulation 2018/213 to approve the restrictions on the use of BPA as food-contacting varnishes and coatings and as a component of plastic FCMs based on Regulation (EU) No 10/2011, reducing the specific migration limit (SML) of BPA from varnishes or coatings applied to FCMs from 0.6 to 0.05 mg/kg of food. However, no migration shall be allowed for varnishes and coatings used in materials and items specifically designed for foodstuffs with direct links to infants and young children in the EU [16]. The current migration limits of BPA in China (0.6 mg/kg) and Japan (2.5 mg/kg) are comparatively higher than in the EU. It is worth noting that France issued a specific regulation in 2012 that suspends the manufacture, import, export, and marketing of any FCMs containing BPA [17] (Table 5).

Furthermore, EFSA established a TDI of 4 µg/kg bw/day of BPA for the population with the highest exposure risk (babies, children, and adolescents) in 2015 [31]. In 2017, a protocol project was developed by an international working group of EFSA under the guidance of a professional group in FCMs of EFSA to re-evaluate the BPA from 2018 [91]. Moreover, on December 15, 2021, the EFSA issued a draft that reduced the TDI of BPA from 4 μg/kg bw/day to 0.04 ng/kg bw/day. It is worth noting that the SML of BPA analogs is 0.05 mg/kg in the EU, China, and South Korea. Therefore, the number of restricted species may increase with the improvement of available information about bisphenol.

## 7. Conclusions and Future Perspectives

In the past decade, considerable advances have occurred in the pathogenic mechanism, migration law, and analytical methodologies for BPA and its analogs in canned foods. However, there were some knowledge gaps in the available literature reviews, requiring urgent attention. There are no systematic and controlled studies comparing the levels of BPA and its analogs in raw materials for the can itself and canned foods. Therefore, it is necessary to quantify a fraction of them originating from the can material used for canned products. Moreover, further research is needed to determine which steps may cause contamination during the production process of raw materials for canned foods. Furthermore, uncertainties and controversies surrounding health impacts are concerning. Therefore, the following studies are required: adverse reactions related to exposure to low doses and combined exposure with other food contaminants; harmful effects of BPA analogs on human health considering the growing and widespread distribution of these new-generation xenoestrogens; different-population for in vivo studies (including vulnerable groups, race, age, eating habits) to clarify the metabolic pathways of these chemicals and to determine internal doses and exposure times related to human exposure.

The remarkable reduction in TDI established by EFSA has tremendously affected the level of risk adjudicated by risk assessments. Consequently, more information on these compound levels and consumption rates of canned products is required. Moreover, using a sufficient sample size is crucial and is recommended to acquire a more accurate evaluation of their concentrations. In addition, responsive packaging systems are expected to achieve considerable advances and massively increase in the next decade to allow real-time food safety and quality monitoring along with remediation (they also have a corrective function) of BPA and its analogs.

It is vital to understand that BPA analogs are being widely used by various industries to replace BPA. Constant monitoring and the threshold of toxicological concern of their presence in canned foods is strongly desirable to increase risk awareness associated with this issue. In addition, a prevention principle must be adopted even if the residual levels of BPA and its analogs are below the SMLs.

## Figures and Tables

**Figure 1 foods-12-01989-f001:**
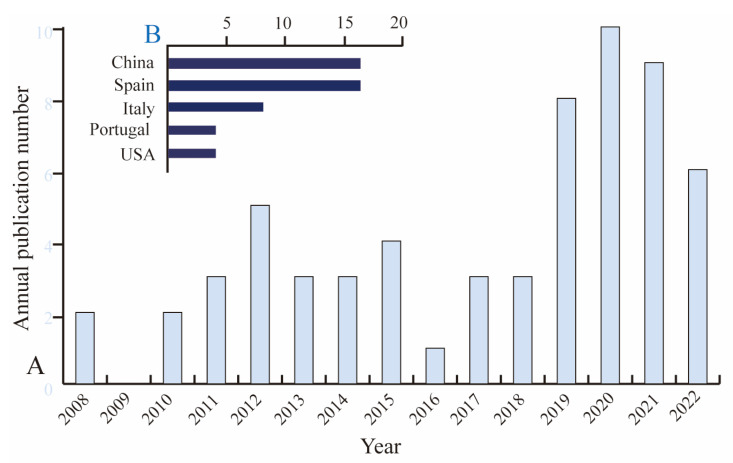
Trend analysis of research on analogs of BPA in canned foods. (**A**) The annual number of publications from 2008 to 2022. (**B**) Top five countries with the highest number of publications from 2008 to 2022.

**Figure 2 foods-12-01989-f002:**
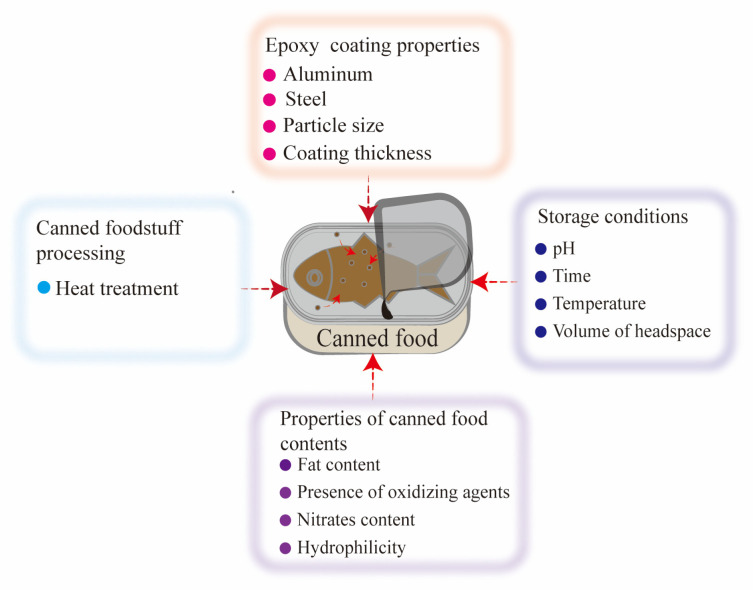
Influencing factors of BPA and its analogs migration from PC-containing coatings to food products contained in cans.

**Figure 3 foods-12-01989-f003:**
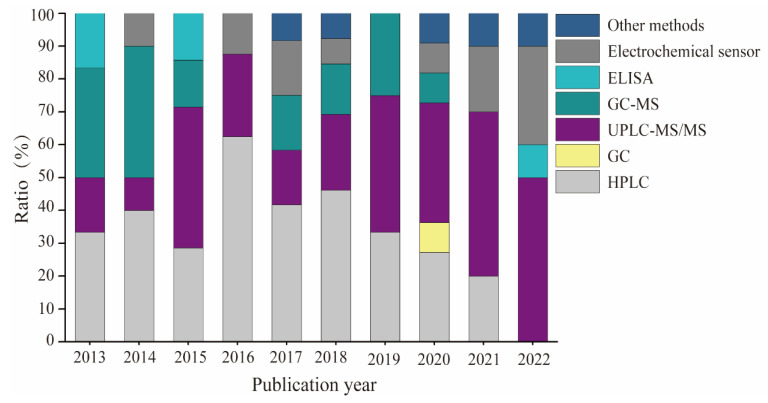
The summary of BPA and its analogs-related determination methods in canned foods in the past decade.

**Table 1 foods-12-01989-t001:** BPA and its analogs included in the systematic review.

No.	Name			Molecular Information		
	Full Name	Abbreviation	Chemical Structure	Molecular Formula	Molecular Weight (g/mol)	CAS
1	Bisphenol A	BPA	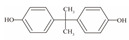	C_15_H_16_O_2_	228.286	80-05-7
2	Bisphenol AF	BPAF	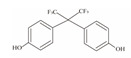	C_15_H_10_F_6_O_2_	336.23	1478-61-1
3	Bisphenol AP	BPAP	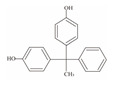	C_20_H_18_O_2_	290.36	1571-75-1
4	Bisphenol B	BPB	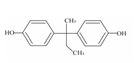	C_16_H_18_O_2_	242.31	77-40-7
5	Bisphenol C	BPC	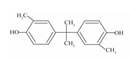	C_17_H_20_O_2_	281.13	79-97-0
6	Bisphenol E	BPE	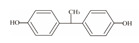	C_14_H_12_O_2_	214.26	2081-08-5
7	Bisphenol F	BPF	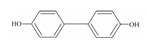	C_13_H_12_O_2_	200.23	620-92-8
8	Bisphenol G	BPG	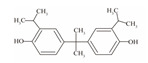	C_21_H_28_O_2_	312.45	127-54-8
9	Bisphenol M	BPM	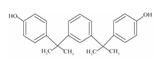	C_24_H_26_O_2_	346.47	13595-25-0
10	Bisphenol P	BPP	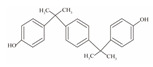	C_24_H_26_O_2_	346.47	2167-51-3
11	Bisphenol PH	BPPH	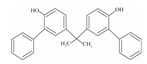	C_27_H_24_O_2_	380.48	24038-68-4
12	Bisphenol S	BPS	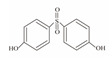	C_12_H_10_O_4_S	250.27	080-09-1
13	Bisphenol TMC	BPTMC	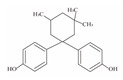	C_21_H_26_O_2_	310.43	129188-99-4
14	Bisphenol Z	BPZ	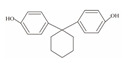	C_18_H_20_O_2_	268.35	843-55-0
15	Bisphenol A Diglycidyl Ether	BADGE	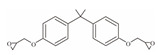	C_21_H_24_O_4_	340.41	1675-54-3
16	Bisphenol F Diglycidyl Ether	BFDGE	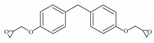	C_19_H_20_O_4_	312.365	2095-03-6

**Table 2 foods-12-01989-t002:** The estimates of the residual levels of BPA and its analogs related to different canned products in recent years.

Sample Origin	Food Matrix	Sample Size	Sampling Time	Compound	Concentration (µg/kg)	Reference
Korea	Canned foods including meats, hams, fish, vegetables, and fruits	104	2015	BPA	1.41–278.50	[41]
				BPF	<0.32	
				BPB	<0.14	
				BADGE·2HCl	1.71–1525.00	
				BADGE·2H_2_O	0.98–655.50	
				BADGE	0.76–27.70	
				BADGE·H_2_O	2.09–24.73	
				BADGE·HCl	1.50–384	
				BADGEHCl·H_2_O	2.42–488	
				BFDGE	<2.85	
China	Canned foods including meats, seafood products, mushrooms, fruits, and vegetables	151	2017–2018	BPA	0.3–837	[48]
				BPF	0.3–75.4	
				BPS	0.05–1.6	
Italy	Canned tuna fish	33	/	BPA	3.7–187.0	[49]
				BPB	3.0–74.8	
				BADGE	7.7–91.1	
				BFDGE	6.9–38.5	
Greek	Canned foods including juices, meats, sauces, fish, and vegetables	14	2014	BPA	0.2–66.0	[50]
Portugal	Canned meats	30	/	BPA	0.15–202.3	[51]
				BPAF	0.4–13.0	
				BPF	0.4–153.0	
				BPB	0.3–33.5	
Spain	Canned foods including fish, seafood, vegetables, grains, and fruit	12	/	BADGE	<0.5	[52]
				BADGE·H_2_O	<0.15	
				BADGE·2H_2_O	0.5–724	
				BADGE·HCl	<0.5	
				BADGE·2HCl	<0.5	
				BADGE·H_2_O·HCl	0.5–189	
Canada	Canned tuna	73	2017–2019	BPA	0.01–29.38	[53]
				BPS	<0.01	
Italy	Canned beers	40	2015	BPA	0.50–0.80	[54]
				BPF	1.1–2.5	
				BADGE	1.2	
				BPB	<0.15	
				BFDGE	<0.15	
Italy	Canned beverages	52	2018	BPA	6.1–76	[55]
				BPB	9.9–183	
				BPE	5.2–59	
				BPF	8.0–139	
				BPM	23–1358	
				BPAF	<5.3	
				BADGE	114	
				BFDGE·2H_2_O	1.21–112.50	
				BFDGE·2HCl	2.15–884.00	
Lebanon	Canned vegetables including fava beans, red beans, chickpeas, and okra	177	/	BPA	12.8–54.6	[27]
				BADGE·2H_2_O	101–146	
				BPZ	1.5–41.0	
Spain	Canned foods including artichokes, asparagus, corn, fruit salad in syrup, green beans, red beans, mackerel, mushrooms, nuts, olive oil, pâté, peach in syrup, squid, and tuna	15	/	BPA	3.45–88.66	[22]
				BPB	0.33–3.86	
				BPE	<0.83	
Thailand	Canned tuna fish	137	2018–2019	BPA	0.195–0.20	[56]

Note: /: Sampling time is not indicated in the reference.

**Table 3 foods-12-01989-t003:** Summarization of the estimated BPA and its analogs exposure doses for the adult population in canned foods from different countries.

Area	The Average Exposure Dose (ng/kg bw/day)		Reference
	BPA	BPA Analogs ^a^	
China	32.9	5.3	[48]
France	22.5	17.15	[50]
Korea	19.33	84.88	[41]
Spain	21.5	156.5	[52]
Spain	370	11	[22]
Texas, USA	12.6	-	[6]

Note: -: The average exposure dose is not indicated in the reference. ^a^: The total exposure amount of all the BPA analogs in the reference.

**Table 4 foods-12-01989-t004:** The cases of the toxicological mechanism of BPA and its analogs.

Adverse Effect	Compound	Subject	Main Discovery	Reference
Cardiotoxicity	BPAF	Zebrafish	Exposure to BPAF increased oxidative stress, inhibited the expression of genes that participate in cardiac development, and played a key role in the mechanism of BPAF-induced cardiac toxicity.	[75]
Cancer	BPA	HeLa cells	BPA increased chromosomal instability by interfering with mitotic processes such as the formation of bipolar spindles and the attachment of spindle microtubules to kinetocomes to stimulate carcinogenic effects.	[65]
	BPS	MCF-7 cells	BPS could change methylation status in the promoter of breast-cancer-related genes.	[66]
	BPA and BPS	MCF-7, MDA-MB-231 breast cancer cells, and SK-BR-3	Stem cell markers and invasion proteins were increased by BPA and BPS in estrogen receptor-positive breast cancer cells.	[76]
Developmental toxicity	BPA	Zebrafish	BPA-induced pharyngeal cartilage defects via cellular pathways such as estrogen receptors, androgen receptors, and estrogen-related receptors.	[77]
	BPA	Zebrafish	The exposure to BPA altered gene expression involved in apoptosis, defense responses, reactive oxygen species metabolism, and signaling pathways in zebrafish larvae and embryos, leading to long-term adverse morphological and functional consequences, including the observed pharyngeal cartilages and craniofacial defects.	[78]
Intestinal toxicity	BPF and BPS	Zebrafish	The individual and combined exposures of BPS and BPF caused oxidative damage and inflammatory effects in the zebrafish intestine. In addition, BPF and BPS exposures changed the microbial community composition in the zebrafish intestine.	[79]
Metabolic disorder	BPA and BPS	Male Wistar rats	The exposure to BPA and BPS elevated the levels of serum lipid markers and upregulated the expression of enzymes involved in triglycerides synthesis. BPS treatment could also enhance liver lipogenic enzyme expressions and have more obesogenic effects compared to BPA.	[69]
	BPS	Male zebrafish	BPS exposure stimulated the expression of autophagy and inflammatory genes, increased the levels of triacylglycerol and total cholesterol in male zebrafish liver, and induced hepatic apoptosis and fibrosis. Long-term exposure to BPS may promote the progression of simple steatosis to nonalcoholic steatohepatitis.	[68]
	BPA	Rats	The exposure to BPA inhibited the mRNA expression of genes encoding insulin leading to poor insulin production. In addition, it significantly reduced glucose uptake via the insulin signaling pathway.	[70]
	BPS	Translucent zebrafish (Danio rerio) larvae	The obesity-inducing effect of BPS is related to the disruption of triacylglycerol metabolism.	[71]
	BPA, BPF, and BPS	Zebrafish embryos/larvae	BPA and its analogs treatment affected several signaling pathways and physiological processes in zebrafish embryos and larvae, especially the metabolic systems.	[80]
	BPS	Zebrafish	BPS exposure significantly reduced the mRNA levels of lipogenic transcription factor srebp1 and increased that of fatty acid synthetase.	[72]
	BPF	Zebrafish larvae	BPF disrupts glucose metabolism and induces hyperglycemia by impairing insulin signaling transduction, increasing the risk of nonalcoholic fatty liver disease in male zebrafish.	[67]
Neurotoxicity	BPB, BPS, BPF, and BPAF	Zebrafish embryos	BPA and its analogs increased the expression of reproductive neuroendocrine-related genes, thereby reducing the body length of zebrafish larvae and affecting their motor behavior.	[80]
	BPA	Human cortical neurons	BPA induced the disorder of intracellular Ca^2+^ homeostasis, thus leading to reactive oxygen species production and anti-oxidative response defect, and then neurite outgrowth disorder, neural network degeneration, and neuron apoptosis.	[73]
	BPA	Rats	Low-dose exposure to BPA disrupted dendritic development and neurotransmitter homeostasis in the hippocampus, leading to impaired spatial learning and memory abilities in rats.	[81]
	BPA	Mice	BPA harmed different memory types and the glutamatergic parameters in a sex-dependent manner.	[82]
Reproductive toxicity	BPA	JEG-3 cells	BPA altered human placental JEG-3 estradiol synthesis and catabolism and interfered with the normal placenta formation process and embryonic development during early pregnancy.	[83]
	BPA	Pregnant women	BPA exposure led to the dysfunction of the hypothalamic pituitary adrenal axis during pregnancy.	[84]
	BPA, BPS, BPAF, and TMBPF	Chicken embryo	BPA and its analogs significantly impaired development, growth, and survival in a dose-dependent manner in the order of BPAF > TMBPF > BPS > BPA. The most common and severe dysmorphologies were body pigmentation abnormalities, craniofacial, eyes, and gastrointestinal.	[85]
	BPS	Zebrafish	BPS has significantly increased 3,5,3′-triiodothyronine plasma levels causing dysontogenesis and incubation delay, bladder inflation defects, decreased motor ability, developmental neurotoxicity, and lateral stripe hypopigmentation in unexposed embryos and larvae.	[86]
Vascular developmental toxicity	BPF, BPS, and BPAF	Zebrafish embryos and human vascular	Exposure to BPA and its analogs increased oxidative stress, including a significant decrease in superoxide dismutase and catalase activity, and increased the levels of malondialdehyde and reactive oxygen species in both zebrafish and human vascular. The order of their vascular toxicity and oxidative stress potency of was as follows: BPAF > BPF > BPA > BPS.	[87]

**Table 5 foods-12-01989-t005:** Specific migration limit values of BPA and its analogs in canned foods for different countries and organizations.

Countries or Organization	Compound	t-TDI (μg/kg bw/day)	SML ^a^ (mg/kg)	Reference
European Union	BPA ^b^	0.04	0.05	[16]
	BPS		0.05	[88]
China	BPA	/	0.6	[89]
	BPS		0.05	
Japan	BPA	/	2.5	[90]
	BPS		0.05	
France	BPA	/	not detect	[17]

Note: TDI (Tolerable Daily Intake) is an estimate of the amount of chemicals that may be harmful to the human body per kilogram of daily intake under long-term contact. t-TDI is adopted if there are uncertainties in the data that can be addressed through further research and it is known that there will be important data updates in the near future. ^a^: SML is the specific migration limit of a single substance based on toxicological assessment according to the TDI established by the Scientific Committee on Food and EFSA. It is supposed that a person weighing 60 kg consumes 1 kg of plastic packaged foods every day throughout their lifetime, which contains the maximum allowable amounts of substances. ^b^: Restriction: BPA should not be used for the manufacture of polycarbonate bottles feeding infants. /: The limit value of t-TDI is not set.

## Data Availability

Data are available from the authors.

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
