# Peer review of "Recent Advances in Sources, Migration, Public Health, and Surveillance of Bisphenol A and Its Structural Analogs in Canned Foods"

_foods, 2023, doi:10.3390/foods12101989_

Round 1
Reviewer 1 Report
The manuscript entitled “Recent Advances in Sources, Migration, Public Health, and Surveillance of Bisphenol A and its Structural Analogs in Canned Foods” comprises a complete review about the occurrence and migration of bisphenol A and its derivatives from canned products to food contained in these containers. In my opinion, the actual review provides a wide enough point of view of the actual situation about the occurrence of these compounds in canned food and highlights the importance of stricter legislation to control them. There are some minor corrections which need to be accomplished.
*Line 13: As you say “canned product” in the next sentence, you could eliminate it in the first sentence
*Line 139: Maybe it could be interesting to make a conclusion with the parameters which increase the BPA migration to have a wider vision of how operate to limit this migration
*Line 171: I would increase this section, mentioning the characteristics of the most employed techniques and the methods allowing the analysis of a wider range of compounds or indicating the number of compounds analysable by each method. For example, I suppose that the analysis with GC implies a derivatization process depending on the analysed compounds.
*Line 218: Clarify how these studies estimated the average exposure levels to BPA from canned products, i.e. by using a migration solvent, analysing the food contained in the cans, etc
*Line 289: Change the title for something simpler as “Bisphenols current legislation”
*Line 338: Change the word “dramatic” or “dramatically”
-
Author Response
Point 1: Line 13: As you say “canned product” in the next sentence, you could eliminate it in the first sentence.
Response 1: The first sentence of line 13 have been eliminated.
Point 2: Line 139: Maybe it could be interesting to make a conclusion with the parameters which increase the BPA migration to have a wider vision of how operate to limit this migration
Response 2: The conclusion of the migration influencing factors have made in Line 213 -221.
Point 3: Line 171: I would increase this section, mentioning the characteristics of the most employed techniques and the methods allowing the analysis of a wider range of compounds or indicating the number of compounds analysable by each method. For example, I suppose that the analysis with GC implies a derivatization process depending on the analysed compounds.
Response 3: The characteristics and the wide range of compounds around the analysis with LC and GC have summarized in line 243 -248.
Point 4: Line 218: Clarify how these studies estimated the average exposure levels to BPA from canned products, i.e. by using a migration solvent, analysing the food contained in the cans, etc
Response 4: It has been clarified in line 287 -288. Namely, the average exposure levels of BPA from canned products regarding with these studies were conducted by analysing the food contained in the cans.
Point 5: Line 289: Change the title for something simpler as “Bisphenols current legislation”
Response 5: The title has been changed to “Bisphenols current legislation” in line 362.
Point 6: Line 338: Change the word “dramatic” or “dramatically”.
Response 6: The words “dramatic” and “dramatically” have been changed in line 412.
Please see the attachment.

Reviewer 2 Report
In this review, the authors present a review of the literature regarding recent advances in the analysis of bisphenol A and analogues in canned foods. The article is very well written and very well structured. The topic is presented in a smooth and clear manner. The article certainly helps to gather and summarize important points for all those interested in this type of issue. Revisions need to be made, however, in what concerns the graphics. Indeed:
Table 1 is graphically unsatisfactory with regard to the chemical structure of bisphenols, which therefore need to be redrawn. The CAS numbers and the full names of bisphenols should not wrap around. I therefore suggest that Table 1 be put in a horizontal format.
The graphics in Figure 1 also need to be revised. In particular, there is unnecessary use of different shades of colors in the graphs shown, making them difficult to read. Monochrome shades are to be preferred.
The graphics in the graphical abstract need to be redone. There is a use of colors, shapes, and designs that is totally inappropriate for the type of article you want to publish. Linear, well-drawn graphics, without unnecessary use of 3D figures, and with professional-looking graphic elements should be preferred. As it stands, figure 2 is not presentable.
Figure 3 is clearer, although we feel that more effort can be made in what concerns the graphics, particularly of the food part.
Author Response
Point 1: Table 1 is graphically unsatisfactory with regard to the chemical structure of bisphenols, which therefore need to be redrawn. The CAS numbers and the full names of bisphenols should not wrap around. I therefore suggest that Table 1 be put in a horizontal format.
Response 1: Table 1 has been put in a horizontal format, and a few of chemical structures have been redrawn.
Point 2: The graphics in Figure 1 also need to be revised. In particular, there is unnecessary use of different shades of colors in the graphs shown, making them difficult to read. Monochrome shades are to be preferred.
Response 2: The figure has been changed using different shade of colors.
Point 3: The graphics in the graphical abstract need to be redone. There is a use of colors, shapes, and designs that is totally inappropriate for the type of article you want to publish. Linear, well-drawn graphics, without unnecessary use of 3D figures, and with professional-looking graphic elements should be preferred. As it stands, figure 2 is not presentable.
Response 3: The use of designs has been changed in the graphical abstract, including changing curved lines to straight lines without using 3D graphics, using professional-looking shapes and appropriate graphic colors.
Point 4: Figure 3 is clearer, although we feel that more effort can be made in what concerns the graphics, particularly of the food part
Response 4: The color of canned food in Figure 3 has been changed.
Please see the attachment.
